# Novel Antimicrobial Peptide from the Hepatopancreas of the Red King Crab

**DOI:** 10.3390/ijms242115607

**Published:** 2023-10-26

**Authors:** Vladislav Molchanov, Alexander Yegorov, Maxim Molchanov, Alexander Timchenko, Vitaly Novikov, Nikolay Novojilov, Maria Timchenko

**Affiliations:** 1Institute of Theoretical and Experimental Biophysics RAS, Pushchino 142290, Russia; vlad-avak@mail.ru (V.M.); egorovae@iteb.pushchino.ru (A.Y.); lvlaks.m@gmail.com (M.M.); 2National Research Nuclear University (NRNU) MEPhI Obninsk Institute for Nuclear Power Engineering (OINPE), Obninsk 249040, Russia; 3Institute of Protein Research RAS, Pushchino 142290, Russia; atim@vega.protres.ru; 4Polar Branch of Russian Federal Research Institute of Fisheries and Oceanography, Murmansk 183038, Russia; nowitaly@yandex.ru; 5M. M. Shemyakin and Yu. A. Ovchinnikov Institute of Bioorganic Chemistry, Russian Academy of Sciences, Moscow 117997, Russia; novojilov_nik93@mail.ru

**Keywords:** antimicrobial peptides, hepatopancreas, red king crab

## Abstract

Crustaceans have successfully adapted to survive in their natural habitat, rich in microorganisms, due to the presence of antimicrobial peptides (AMPs) in their organism. They achieve this adaptation despite lacking the highly specific adaptive immune system found in vertebrates. One valuable source of AMPs is the hepatopancreas, a waste product from crab fishery and its processing. Applying zymographic and spectrophotometric techniques, we discovered a small peptide (approximately 5 kDa) within a low molecular weight protein fraction extracted from the acetone powder of the red king crab hepatopancreas. This peptide hydrolyzes both *M. lysodeikticus* cell wall and *M. lysodeikticus* cell wall polysaccharide, while showing no activity against gelatin. The found peptide may be of interest for application in medicine, biotechnology, and the food industry, for example as a bio-preservative.

## 1. Introduction

Natural antimicrobial peptides (AMPs) are small peptides with a molecular weight ranging from 2 to 9 kDa [1]. AMPs are evolutionarily ancient factors of the humoral innate immune system and play a key role in immune response, providing protection against a wide range of viruses, fungi, and bacteria, disrupting the structure or function of the cell membrane in the latter. AMPs are found in all kingdoms, from bacteria to mammals, including plants. To date, there is growing evidence that AMPs also play a crucial role in human immunity. Approximately a hundred AMPs have been identified in barrier epithelial tissues, phagocytic cells, and human biological fluids [2,3]. At the moment, more than 3 thousand natural AMPs have been discovered (https://aps.unmc.edu/AP (accessed on 14 August 2023)), and a third of them are invertebrates [1].

Antimicrobial activity in hemolymph or hemocytes as well as in other organs is found in several decapod crustaceans, including lobsters, crabs, shrimps, and freshwater crayfish [4,5,6,7]. The powerful antimicrobial activity in crustaceans is due to their constant exposure to various pathogens, such as viruses, bacteria, fungi, and other parasites, and the absence of specialized lymphoid organs in their immune system. Crustacean defense mechanisms depend entirely on the innate immune system, whose major molecular factors are antimicrobial proteins and peptides. The red king crab (*Paralithodes camtschaticus*) is the best-known and most widespread member of the *Lithodidae* (*Anomura*) family of the crustacean order *Decapoda*. To date, the full genome sequence of the red king crab is unknown, and there are not many works devoted to the study of enzymes of the crab from the infraorder *Anomura* (craboids), to which it belongs [8]. The individual studies devoted to newly discovered enzymes from the hepatopancreas of the red king crab demonstrate that the crab fishery’s waste byproduct, the hepatopancreas, is a source of valuable enzymes with high activity and a wide range of action. These enzymes possess unique properties distinct from previously described enzymes of other crustaceans [9,10]. The hepatopancreas of the red king crab can also be considered as a source of new AMPs, which could become prototypes of new candidates for broad-spectrum antibiotics and take a step towards solving the problem of antibiotic resistance, which poses a serious global threat.

In this study, extracts of low-molecular-weight protein fractions were obtained from the acetone powder of the red king crab hepatopancreas in two different ways, and their activity toward the cell wall of the gram-positive bacterium *M. lysodeikticus* and the polysaccharide of the cell wall of *M. lysodeikticus* was studied using zymography, spectrophotometry, and NMR spectroscopy. Moreover, the effect of these samples on the growth of certain gram-positive bacteria (*B. cereus* and *B. subtilis*) and gram-negative microorganisms (*E. coli*) was studied.

## 2. Results

### 2.1. Analysis of the Activity of Extracts of Low-Molecular-Weight Protein Fractions from the Hepatopancreas of the Red King Crab toward the Cell Wall of the Gram-Positive Bacterium M. lysodeikticus

To identify proteins and peptides with antibacterial activity, low molecular weight protein extracts were isolated from the acetone powder of the king crab hepatopancreas (HPC). The extracts were prepared using two methods to assess the effectiveness of each approach. The first method involved the extraction of a low-molecular-weight protein fraction with 60% acetonitrile (ACN) and 0.1% trifluoracetic acid (TFA) [11], and the second one involved extraction with a solution of 2 mM dithiothreitol (DTT) and 0.1% formic acid [12]. The latter was proposed by Fukutomi et al. [12] for the quantitative purification of peptides from crude extracts of animal tissues because they are more soluble in DTT than large proteins. Initially, the obtained extracts were analyzed for activity toward the cell wall of the gram-positive bacterium *M. lysodeikticus* by zymography. As a positive control, a hen egg-white lysozyme (HEWL) (3 mg/mL) was used. HEWL is a muramidase that hydrolyzes cell wall polysaccharides. The samples were loaded into a polyacrylamide gel containing a purified cell wall (~1 mg/mL), and SDS-PAGE was performed. Activity was determined by the appearance of white bands on the zymogram after renaturation of proteins in gel by incubation with renaturation buffer (pH 7.2) for 2 h at 37 °C and methylene blue staining. White bands indicated the position where the *M. lysodeikticus* cell wall was hydrolyzed by cell wall hydrolases (Figure 1). The second half of the gel was stained with Coomassie to determine the size and location of the proteins of interest. To determine the molecular weight, protein markers with known molecular weights were used (Figure 1).

It was found that one individual substrate cleavage band appeared in the zymogram in the range of 5 kDa for both extracts obtained by two different methods. This indicates the presence of a cell wall-hydrolyzing enzyme in the studied samples. The low intensity is due entirely to the small amount of the applied samples. The activities of samples obtained by different methods are comparable.

### 2.2. Analysis of the Activity Type of Extracts of Low-Molecular-Protein Fractions from the Hepatopancreas of the Red King Crab

Bacterial cell wall digestion occurs through two main pathways: one involves the polysaccharide part (hydrolase, including muramidase activity), and the other involves peptide fragments (protease activity). To determine whether the activity of the found enzyme toward the cell wall is associated with the destruction of its peptide part, the gelatin-hydrolyzing activity of the extracts of low-molecular-weight protein fractions obtained in the two different aforementioned ways was studied. Proteinase K and the supernatant obtained after dissolving the initial HPC acetone powder in 25 mM phosphate buffer, pH 5.5, followed by centrifugation were used as positive controls (Figure 2).

Figure 2 reveals that the studied extracts did not show gelatinase activity in the 5 kDa region (no white band). This suggests that the enzyme in the extracts that is acting on the cell wall is not hydrolyzing it through the peptide fragments. In the case of the first HPC sample, strong gelatinase activity was detected at 25 kDa. Both the initial HPC sample and the studied extracts also exhibited weak gelatinase activity bands in the 25–30 kDa range as well as at 40 kDa. To understand whether the digestion of the cell wall is associated with the cleavage of the cell wall polysaccharide and the hydrolase activity of the found 5 kDa protein, zymography with the *M. lysodeikticus* cell wall polysaccharide as a substrate was performed (Figure 3). HEWL lysozyme was used as a positive control for hydrolase activity.

Figure 3 demonstrates that the protein in the extracts effectively cleaves the polysaccharide of the cell wall from the gram-positive bacterium *M. lysodeikticus*, confirming the presence of hydrolase activity. The obtained results were confirmed by NMR analysis data.

Proton NMR spectra of the initial polysaccharide, the extract of low molecular weight protein fraction isolated by the acetonitrile method in the absence of a polysaccharide, and a polysaccharide after a five-day incubation at 37 °C with the peptide extract were obtained (Figure 4).

Figure 4 represents the sum of the individual spectra (individual polysaccharide and peptide extract) as well as the spectrum of the reaction mixture. As can be seen from the NMR spectra (Figure 4), after incubation with the peptide extract the spectrum of the polysaccharide changed significantly in the region of bound acetyl groups (Figure 4a) and the region of amino groups (Figure 4b). An increase in signals in the reaction mixture compared to the initial polysaccharide indicates the hydrolysis of the substrate and the appearance of shorter polysaccharide molecules [9]. In the absence of polysaccharide, no signals appear in these regions. Therefore, NMR spectroscopy confirmed the hydrolase activity in the peptide extract. A comparison of the NMR spectra of the products of polysaccharide hydrolysis over three days with peptide extracts obtained using ACN and DTT was also carried out (Figure 5). The obtained results show that neither the activity nor the mechanism of action of extracts obtained by various methods differs fundamentally.

The peptide extract itself did not give significant signals in the studied spectral region (Figure 6).

To analyze the resistance of the polysaccharide itself to incubation at 37 °C, spectra of the polysaccharide were obtained before incubation and after five days of incubation at 37 °C. No differences were observed between the spectra (Figure 7).

To exclude the contribution of other protein molecules that may be present in the extract, a peptide preparation was prepared by excising the activity spot from the zymogram, eluting the peptide from the gel using 40% ACN with 0.1% TFA [12], and freeze-drying the sample. The sample was then dissolved in 50 μL of 25 mM phosphate buffer pH 5.5 and incubated with the polysaccharide for 5 days in the same way as the extract, followed by NMR analysis. Figure 8 shows that although the changes are not as pronounced due to the low concentration as for the extract, hydrolysis of the polysaccharide occurs, which is manifested by an increase in the signal due to the appearance of shorter polysaccharide chains.

### 2.3. Analysis of the Activity of Extracts of Low-Molecular-Weight Protein Fractions from the Hepatopancreas of the Red King Crab at Different pH

The pH range at which the enzymes are active against the cell walls was determined by zymography. This included renaturation in buffers of different pH and spectrophotometric analysis of the change in absorbance at 540 nm of the cell wall suspension incubated with the extract at different pH. Acetonitrile peptide extract was used for this analysis. Zymography was carried out according to the method described above. Following gel electrophoresis, the gel was divided into three parts and then renatured in three different buffers: 25 mM sodium phosphate buffer pH 5.5, 25 mM Tris-HCl (pH 7.2), and 25 mM Tris-HCl (pH 9.0) containing 1% (*v*/*v*) Triton X-100 (Figure 9).

It was found (Figure 9) that the digestion at pH 9.0 by the peptide was slightly worse. The data are also confirmed by spectrophotometry. Figure 10a demonstrates a comparison of the absorption of a cell wall suspension at 540 nm, both in the absence of enzyme and after incubation at 37 °C with an extract of a low molecular weight protein fraction from the hepatopancreas of red king crab obtained using acetonitrile. This figure indicates that the extract exhibits maximum activity at pH 7.2 and the lowest activity at pH 9.0. HEWL lysozyme served as a control and exhibited a similar trend, as shown in Figure 10b.

### 2.4. Analysis of the Effect of Extracts of Low-Molecular Protein Fractions from the Hepatopancreas of the Red King Crab on Bacterial Growth

To evaluate the antibacterial activity of the obtained extract, the growth of cultures of gram-positive microorganisms *B. cereus* and *B. subtilis*, as well as gram-negative *E. coli*, was tested in the presence of an acetonitrile-extracted peptide fraction. The assessment was carried out spectrophotometrically by measuring the changes in the optical density of the cell cultures at 600 nm [11,12]. HEWL lysozyme was used as a positive control for antibacterial activity (Figure 11).

The growth of cultures clearly demonstrates the peptide extract’s ability to suppress gram-positive bacteria growth and slightly reduce gram-negative bacteria growth at the initial stage. The decrease in the growth of the culture (Batch bioprocess) without antibacterial agents is most likely due to the fact that the culture at such a high density does not have enough nutrients, the aeration is also pure, and some metabolites can inhibit growth and result in cell death. The presence of antibacterial agents prevents such a high culture density. Therefore, low molecular weight protein fractions from acetone powder of the king crab hepatopancreas likely exhibit antibacterial activity due to the presence of a protein with a molecular weight of about 5 kDa, effectively hydrolyzing the bacterial cell walls. This discovered activity, particularly against bacteria such as *B. cereus*, the most common cause of toxic infections, already suggests that the obtained extracts could serve as preservatives for food storage.

## 3. Discussion

Every year, the problem of antibiotic resistance becomes more acute, as evidenced by the increase in nosocomial infections, which are responsible for 50% of all hospitalization complications [13]. The search for new molecules with antibacterial properties is constantly underway, because they could form the basis for the development of new antibiotics. The perspectiveness of AMPs is explained by a number of reasons [13]: (1) the small size of AMPs (<40 residues), (2) their systemic action, (3) high affinity and wide spectrum of activity toward microbial cell membranes, (4) resistance to proteolysis, (5) rapid suppression of bacteria and fungi growth, (6) limited immunogenicity, and (7) lack of bacterial resistance to them. Marine organisms capable of surviving under conditions of high salinity, pressure, pH, and temperature, often without a highly developed immune system, are of particular interest in the search for new AMPs. AMPs from marine organisms are characterized by unique amino acid sequences and various post-translational modifications [14]. A promising source for discovering bioactive peptides are marine invertebrates that can survive a wide temperature range. To date, 15 families of AMPs or individual peptides have been identified in crustaceans, such as penaeidins, tachyplesins, and polyphemusins, which share common molecular characteristics with known AMP families [15]. One of the most abundant sources of AMPs can be found in the hepatopancreas of the red king crab, a byproduct of crab fishing. Previously, paralythocins were discovered in the hemocytes of the red king crab [16], and the sequence of crustin was determined [17]. Haug et al. [11] also identified antibacterial activity in the hepatopancreas of the king crab but did not specify the source of this activity.

To investigate potential antimicrobial peptides in the red king crab hepatopancreas, we obtained extracts of low-molecular-weight protein fractions from acetone powder of the red king crab hepatopancreas and evaluated their activity toward the cell wall of the gram-positive bacterium *M. lysodeikticus* and the cell wall polysaccharide of *M. lysodeikticus*. Their effect on the growth of strains of a number of microorganisms (*B. cereus*, *B. subtilis*, and *E. coli*) was also studied. The extracts were obtained by two methods: the most commonly used method using 60% acetonitrile and 0.1% TFA [11], and the method proposed by Fukutomi T. [12]. In Fukutomi’s method, peptides free of high-molecular-weight proteins were obtained by dissolving acetone-precipitated tissue extracts in 2 mM DTT with 0.1% formic acid. Peptides and low-molecular-weight proteins have higher solubility in 2 mM DTT than high-molecular-weight proteins. Zymographic analysis showed that the extracts analyzed from the hepatopancreas of the red king crab contained a small protein with a molecular mass of about 5 kDa, which appeared as a single band on electrophoresis. This protein hydrolyzes both the cell wall of *M. lysodeikticus* and the cell wall polysaccharide composed of N-acetylmuramic acid and N-acetylglucosamine, while showing no activity against gelatin. This observation indicates that the detected peptide possesses hydrolase activity. The cleavage of the polysaccharide was also confirmed by ^1^H NMR spectroscopy. It should be noted that extracts obtained with both ACN and DTT showed almost comparable activity. The most efficient hydrolysis of the cell wall was observed at neutral pH and the worst at alkaline pH. Preliminary analysis of the antibacterial activity of the obtained extracts against the gram-negative bacterial strain *E. coli* and the gram-positive *B. cereus* and *B. subtilis* showed that the addition of the peptide extract inhibited the growth of *B. cereus* and *B. subtilis*, and the growth of *E. coli* was retarded at the initial growth stage.

Thus, in the hepatopancreas of the king crab there is a peptide that exhibits antibacterial activity by cleaving cell wall peptidoglycan at glycosidic bonds, similar to lysozyme. As is well known, the major known AMPs can be divided into those acting by a membrane-targeting mechanism and those acting by non-membrane targeting mechanism according to their mechanism of action. The first is the electrostatic interaction of peptides with the bacterial membrane and the further formation of pores in it according to the carpet model, barrel-stave model, toroidal pore model, as well as the formation of micellar complex according to the agglutination model resulting in the aggregation of bacterial cells. This leads to the killing of the pathogens. According to the non-membrane targeting mechanism, AMPs can inhibit the synthesis of proteins, nucleic acids, lipoteichoic acid, and peptidoglycan; interact with proteins; or affect the replication and synthesis of DNA and RNA [18]. At the same time, AMP from the king crab hepatopancreas acts enzymatically on the bacterial membrane and destroys it. It cannot be excluded that the activity found is caused not only by AMPs from the king crab hepatopancreas but also by AMPs from symbiotic bacteria that may be present in the hepatopancreas. It is known that bacteria that produce bacteriocins, such as enterococci, can be found as symbionts in the intestines of animals. Bacteriolysins (formerly class III bacteriocins) function by lysing sensitive cells by catalyzing cell wall hydrolysis [14]. It has also been reported that microorganisms may be associated with the microvilli boundary in the hepatopancreas of terrestrial isopods [19].

Therefore, to determine the nature of the peptide found, the study requires further mass spectrometric analysis and peptide identification. Obtaining a pure peptide in sufficient quantity allows not only the evaluation of peptide activity against a wide range of microorganisms, but also the determination its structure using NMR spectroscopy.

Further identification of this peptide from the red king crab could potentially be important in expanding the list of known AMPs and solving the problem of bacterial resistance to antimicrobial drugs.

## 4. Materials and Methods


*Preparation of acetone powder from red king crab hepatopancreas.*


Acetone powder from red king crab hepatopancreas was prepared by mincing frozen hepatopancreas, mixing it with acetone (1:8 ratio), and keeping it in a freezing chamber for 15–20 h with periodic stirring. The suspension was filtered, and the extraction was repeated with cooled acetone and n-butanol. The protein precipitate was dried in a vacuum desiccator (Medfizpribor, Kazan, Russia) and a lyophilic dryer Heto FD 8 (Heto-holten A/s, Allerød, Denmark). For the extraction of lipid substances, “chemically pure” acetone (Panreac, Barcelona, Spain) and “pure” n-butanol (Acros Organics, Geel, Belgium) were used.


*Extraction of low molecular weight protein fractions from acetone powder of red king crab hepatopancreas.*


Samples of peptide extracts from acetone powder of red king crab hepatopancreas were obtained by two methods.

Method 1 [11]:

Freeze-dried samples of acetone powder from red king crab hepatopancreas (~100 mg) were extracted with 10 volumes (*v*/*w*) of 60% (*v*/*v*) acetonitrile (ACN; HPLC grade, Panreac, Barcelona, Spain) containing 0.1% trifluoroacetic acid (TFA, AppliChem, Darmstadt, Germany) for 24 h at 4 °C. The supernatant was collected, stored at 4 °C, and the residue was re-extracted under the same conditions. The combined supernatants were incubated at 20 °C for 1–2 h to form two liquid phases: an acetonitrile-rich phase and an aqueous-rich phase. The aqueous extract (bottom phase) was lyophilized and dissolved in 50 μL of sodium phosphate buffer pH 5.5 (50 mM Na_2_HPO_4_—NaH_2_PO_4_ (pH 5.5)).

Method 2:

The second method is based on the preparation of peptides for mass spectrometric analysis as described in [12]. Acetone powder lyophilizate of red king crab hepatopancreas was dissolved in 2 mM dithiothreitol (DTT, Sigma Aldrich, St. Louis, MO, USA) containing 0.1% formic acid with stirring. The solution was centrifuged at 10,000× *g* for 15 min at 4 °C, and the peptide-enriched supernatant was collected, as low-molecular-weight peptides and proteins are more soluble in 2 mM DTT than high-molecular-,weight ones. The peptide-enriched supernatant was lyophilized and dissolved in 50 μL of pH 5.5 sodium phosphate buffer.


*Preparation of cell wall of M. lysodeikticus and the polysaccharide of cell wall of M. lysodeikticus.*


The cell wall of *M. lysodeikticus* and the polysaccharide that was isolated from it were provided by the Experienced Biotechnological Plant (EBP) of IBCh RAS. *M. lysodeikticus* cell wall preparation was obtained by mechanical cell disruption, cell wall separation on centrifuge, washing, and drying. The cell wall polysaccharide solution was prepared from the cell wall preparation of *M. lysodeikticus* by its enzymatic treatment with amidase (Sigma Aldrich, St. Louis, MO, USA) and one step of chromatographic purification. The amount of polysaccharide composed of N-acetyl muramic acid and N-acetyl glucosamine was measured by the anthrone method for the determination of reducing carbohydrates (by glucose) [20]. The sample contained polysaccharide at a concentration of 3.3 mg/mL in the following buffer: 20 mM sodium acetate, 0.5 M NaCl, pH 5.5.


*Polyacrylamide gel zymography with sodium dodecyl sulfate (SDS-PAGE).*


(A)Zymography for detection of hydrolase activity

Zymography to detect hydrolases active toward cell wall and cell wall polysaccharide was performed according to the protocol [21].

The separation gel contained *M. lysodeikticus* cell wall (10 mg purified cell wall per 10 mL) or *M. lysodeikticus* cell wall polysaccharide (0.5 mg/mL). After preparation of the SDS-PAGE gel, the tested samples were loaded onto the gel in the loading buffer and electrophoresis was performed [22]. Hen egg-white (HEW) lysozyme (HEWL) (0.5 mg/mL) (Amresco, Solon, OH, USA) was used as a positive control. The gel was cut in two pieces, one for Coomassie protein staining and the other for renaturation and further activity detection using methylene blue staining (0.01% (*w*/*v*) methylene blue and 0.01% (*w*/*v*) KOH in distilled water). The renaturation was performed in 25 mM Tris-HCl buffer (pH 7.2) containing 1% (*v*/*v*) Triton X-100 for 2 h at 37 °C, followed by staining with methylene blue for 20 min followed by washing with distilled water for 12 h. The Coomassie R-250 staining was performed according to [21]. To detect activity at other pH values, 25 mM sodium phosphate buffer pH 5.5 and 25 mM Tris-HCl (pH 9.0) containing 1% (*v*/*v*) Triton X-100 were used as a renaturation buffer. The presence of hydrolase activity was assessed by the appearance of white bands in the blue background gel.

Later, cell wall zymography was used to test the activity of low-molecular-weight protein fractions each time the extracts were isolated.

(B)Zymography for detection of gelatinase activity

Zymography was performed according to the protocol [23]. The separating and concentrating gels were prepared in the same manner as for the hydrolase activity zymography, except that the separating gel contained 0.1% gelatin (1 mg/mL) as a substrate (Sigma Aldrich, St. Louis, MO, USA). After two 30 min washings in 2.5% (*v*/*v*) Triton X-100, renaturation was performed in 50 mM Tris-HCl buffer (pH 7.5) containing 1% (*v*/*v*) Triton X-100, 5 mM CaCl_2_, and 1 μM ZnCl_2_ for 24 h at 37 °C, followed by 1 h of Coomassie R-250 staining and gel washing. The appearance of white bands in the blue background gel was used to assess the presence of gelatinase activity. Proteinase K (Sigma Aldrich, St. Louis, MO, USA) and the supernatant obtained after dissolution of the initial HPC acetone powder in 25 mM phosphate buffer, pH 5.5, and centrifugation at 10,000× *g* for 15 min were used as controls.


*Elution of HPC peptide from gel.*


White band in a sample lane of the SDS-PAGE gel was excised after zymography, and the piece of gel was placed in an Eppendorf. The peptide was extracted by shaking in 700 μL of 40% acetonitrile containing 0.1%TFA for 16 h. The gel piece was further washed for 8 h and briefly with 300 μL and with 100 μL, respectively, of the same solution. The three washes were pooled, and the gel fragments were removed by passage through a 0.45 μm-pore filter (Millipore, Burlington, MA, USA). The filtrate was lyophilized and dissolved in 50 μL of 25 mM phosphate buffer, pH 5.5


*Turbidimetric analysis of the activity of peptide samples toward M. lysodeikticus cell wall.*


The absorbance of the cell wall suspension at 540 nm (A_540_ = 1) corresponds to approximately 1 mg of cell wall per milliliter [21]. The purified cell wall (0.99 OD_540_) was resuspended in 3 mL of the buffer required for the assay. The final absorbance of the suspension at 540 nm was 0.33 OD_540_. The protein extract (4 mg lyophilized extract per 1 mL) or lysozyme (0.02 mg/mL) were added to 3 mL of the cell wall suspension as controls, and the samples were incubated at 37 °C. The absorbance at 540 nm was measured at certain time intervals during the incubation.


*Analysis of hydrolase activity of peptide samples towards M. lysodeikticus cell wall polysaccharide by NMR spectroscopy.*


A.Sample preparation

The following samples were prepared for evaluation of hydrolase activity (total sample volume—570 µL): controls—polysaccharide (180 µL polysaccharide, 390 µL 25 mM sodium phosphate buffer pH 5.5); low-molecular-weight protein fraction extract obtained by acetonitrile extraction (1 method) (10 µL peptide extract, 560 µL 25 mM sodium phosphate buffer pH 5.5); and reaction mixtures containing 50 µL polysaccharide, 10 µL low molecular weight protein fraction extract, and 510 µL 25 mM sodium phosphate buffer pH 5.5. The extracts obtained with acetonitrile (method 1) and dithiothreitol (method 2) were used. The controls and reaction mixtures were incubated for three days at 37 °C. For acetonitrile extract, an additional experiment was performed with samples incubated for 5 days at 37 °C. Prior to NMR analysis, 30 μL of the standard, a 4 mM solution of 3-trimethylsilyl-[2,2,3,3,3-^2^H_4_] sodium propionate (TSP, Sigma Aldrich, St. Louis, MO, USA) in 1 M phosphate buffer (pH 7.2) containing D_2_O (Sigma Aldrich, St. Louis, MO, USA), was added to 570 μL of the samples.

B.Detection of hydrolase activity

The samples (600 µL) were placed into an NMR tube with a diameter of 5 mm (Wilmad Labglass, Vineland, NJ, USA).

The 1D spectra were recorded on the Bruker 600 AVANCE III NMR spectrometer (Bruker Biospin, Rheinstetten, Germany), operating at a frequency of 600 MHz for protons, using standard pulse sequences from the Bruker pulse sequence library. All measurements were carried out at a temperature of 298 K (25 °C). To suppress the signal from water protons, a pre-saturation method was used by applying a 1D pulse sequence ZGPR. The number of accumulations ranged from 64 to 128 scans, and the interval between scans was 10 s. The chemical shifts were assigned according to the TSP signal at 0.00 ppm, which acts as an internal reference. The spectrum processing and the calculation of the integrals were carried out by the Bruker company «TOPSPIN» program (Bruker Biospin, Rheinstetten, Germany). The spectral database of Bruker’s AMIX software (Bruker Biospin, Rheinstetten, Germany) was used to validate our results.


*Testing of antibacterial activity on cell cultures.*


Aliquots of 50 μL each of the test fractions (peptide extract obtained with acetonitrile and lysozyme (3 mg/mL) as positive control) were incubated with 5 mL of a medium-log phase bacterial culture suspension (gram-positive bacteria *B. cereus* (ATCC 4342), *B. subtilis* (MoBiTec WB800N), and gram-negative bacterium *E. coli* (ATCC 25922)). Strains were provided from the All-Russian Collection of Microorganisms of IBPM RAS. The peptide extract or lysozyme for the antibacterial activity assay were added after reaching the optical density of 0.150–0.200 OD at 600 nm [11]. The incubation was performed at 37 °C with constant shaking. The bacterial growth was evaluated by measuring the optical density at 600 nm after the addition of extract samples at 6, 16, 22, and 26 h for *B. cereus* and 4, 14, 20, and 24 h for *E. coli*; the bacterial growth of *B. subtilis* in the presence of extracts at 6, 19, and 25 h was also evaluated. The negative controls included medium with sample added and pure medium with Milli-Q added in the same volume as the sample.


*Statistical analysis.*


The experiments were repeated, independently performing the process of obtaining extracts and further analysis, more than three times. The results of three separate experiments were averaged for the turbidimetric analysis data presented. Error bars represent standard deviation.

## Figures and Tables

**Figure 1 ijms-24-15607-f001:**
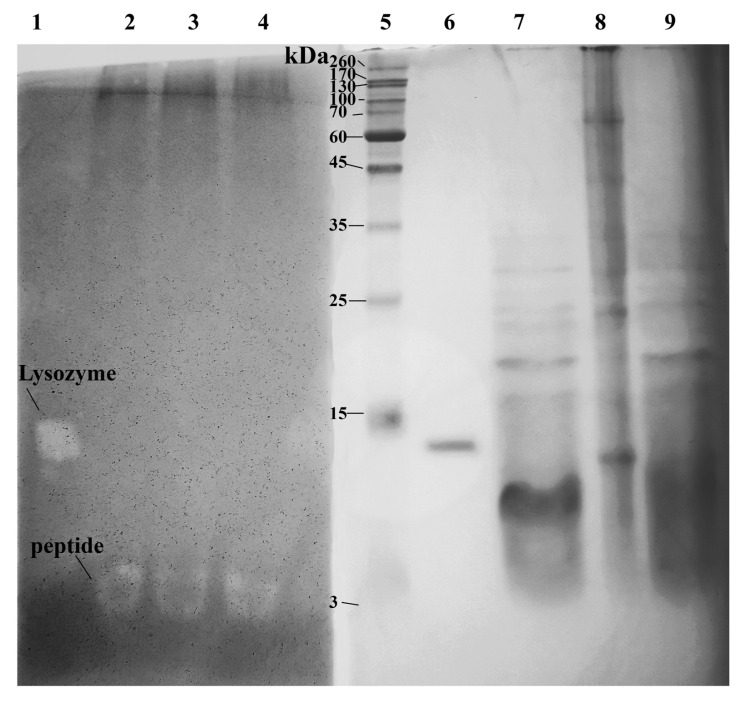
Study of the activity of extracts from acetone powder of the red king crab hepatopancreas toward the *M. lysodeikticus* cell wall by zymography. Lanes 1–4—methylene blue staining, lanes 5–9—Coomassie staining. Lanes: 1, 6—HEWL lysozyme; 2, 4, 7, 9—HPC extract obtained with ACN; 3, 8—HPC extract obtained with DTT; 5—Molecular weight markers.

**Figure 2 ijms-24-15607-f002:**
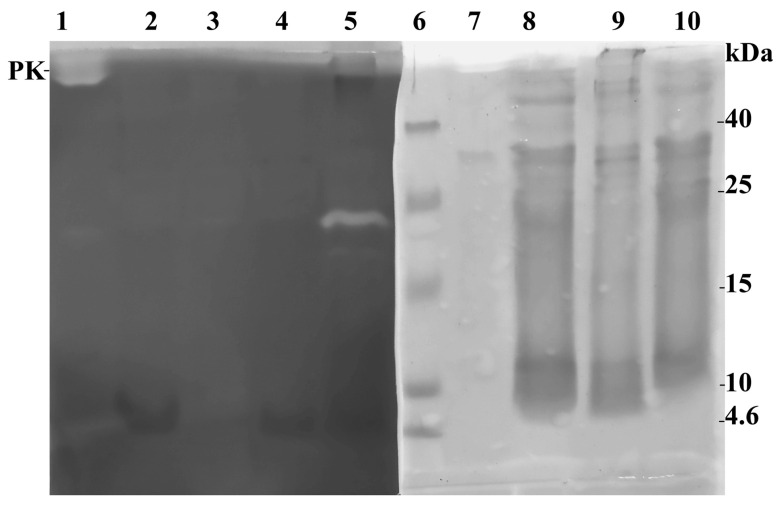
Study of the gelatinase activity of extracts from acetone powder of the red king crab hepatopancreas by zymography. Lanes: 1, 7—proteinase K (PK); 2, 4, 8, 10—HPC extract obtained with ACN; 3, 9—HPC extract obtained with DTT; 5—extract of the initial HPC acetone powder in 25 mM phosphate buffer, pH 5.5; 6—Molecular weight markers.

**Figure 3 ijms-24-15607-f003:**
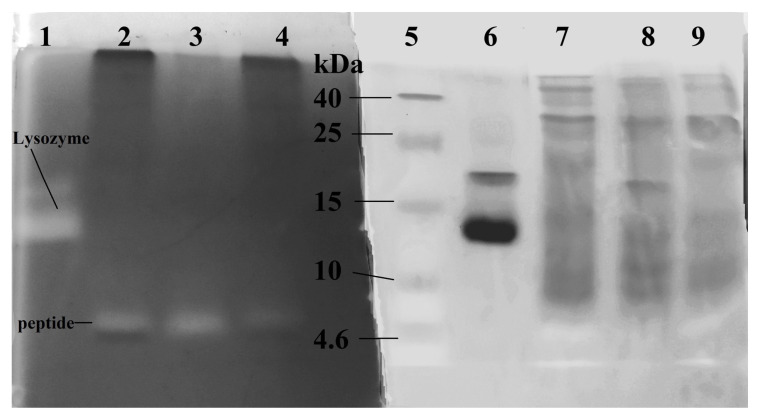
Study of the activity of extracts from acetone powder of the red king crab hepatopancreas toward the *M. lysodeikticus* cell wall polysaccharide by zymography. Lanes 1–4—methylene blue staining, lanes 5–9—Coomassie staining. Lanes: 1, 6—HEWL lysozyme; 2, 4, 7, 9—HPC extract obtained with ACN; 3, 8—HPC extract obtained with DTT; 5—Molecular weight markers.

**Figure 4 ijms-24-15607-f004:**
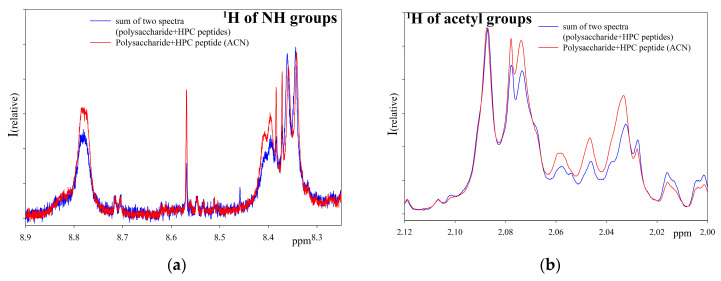
^1^H NMR spectra of *M. lysodeikticus* cell wall polysaccharide hydrolyzates prepared by peptide extract from acetone powder of the red king crab hepatopancreas at pH 5.5 ((**a**,**b**) show different regions of the spectrum where changes in the signals of the initial polysaccharide during cleavage are observed).

**Figure 5 ijms-24-15607-f005:**
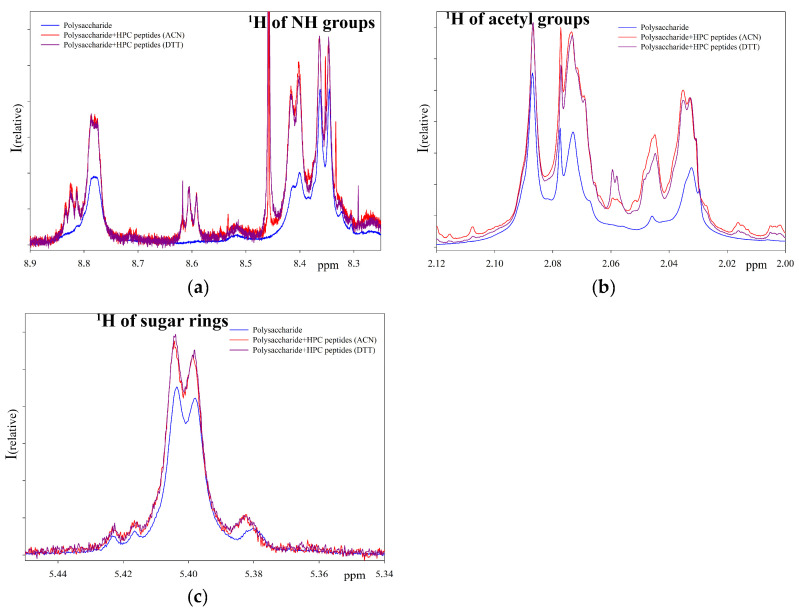
^1^H NMR spectra of *M. lysodeikticus* cell wall polysaccharide hydrolysis at pH 5.5 with peptide extracts from acetone powder of the red king crab hepatopancreas, obtained using ACN and DTT ((**a**–**c**) show different regions of the spectrum in which changes in the signals of the initial polysaccharide are observed upon hydrolysis).

**Figure 6 ijms-24-15607-f006:**
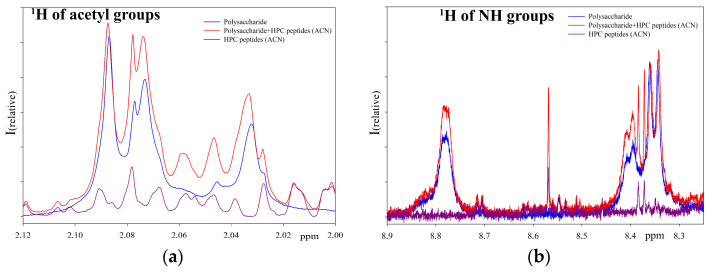
^1^H NMR spectra of *M. lysodeikticus* cell wall polysaccharide hydrolysis at pH 5.5 with peptide extracts from acetone powder of the red king crab hepatopancreas, obtained using ACN (**a**,**b**) show different regions of the spectrum in which changes in the signals of the initial polysaccharide are observed upon hydrolysis).

**Figure 7 ijms-24-15607-f007:**
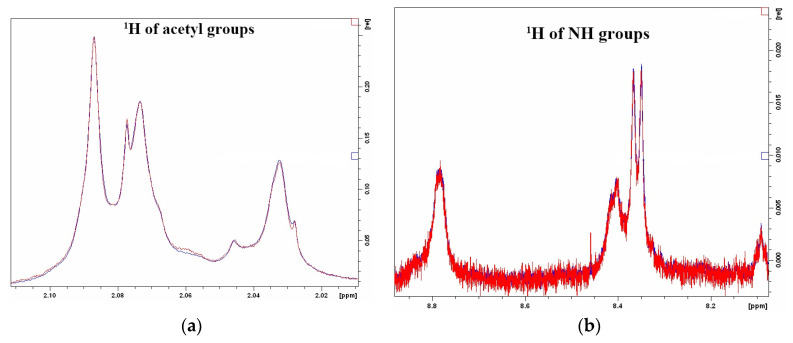
^1^H NMR spectra of *M. lysodeikticus* cell wall polysaccharide before and after incubation at 37 °C ((**a**,**b**) show different regions of the spectrum where hydrolysis of the polysaccharide was previously observed in the presence of the peptide).

**Figure 8 ijms-24-15607-f008:**
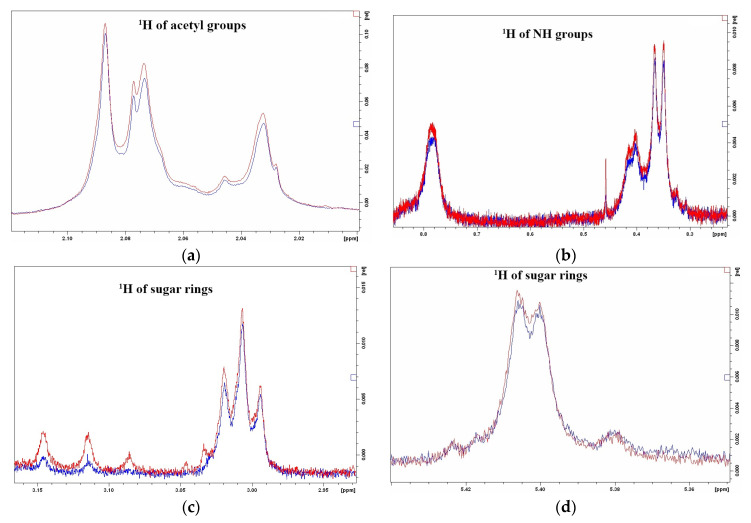
^1^H NMR spectra of *M. lysodeikticus* cell wall polysaccharide hydrolysis at pH 5.5 with eluated peptide, obtained using ACN ((**a**–**d**) show different regions of the spectrum in which changes in the signals of the initial polysaccharide are observed upon hydrolysis).

**Figure 9 ijms-24-15607-f009:**
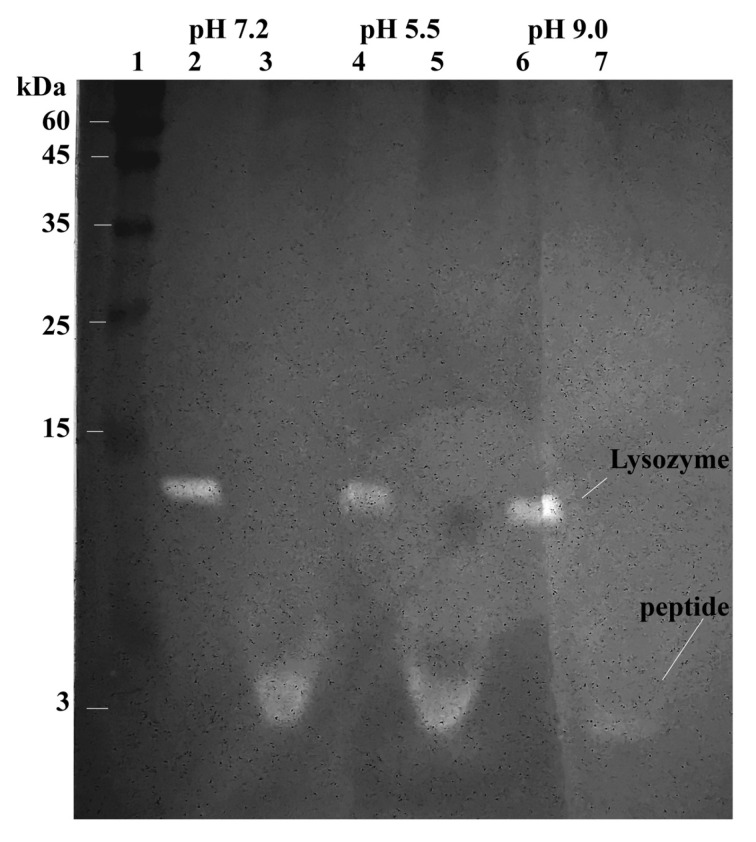
Study of the activity of extracts from acetone powder of the red king crab hepatopancreas toward the *M. lysodeikticus* cell wall at different pH by zymography. Lanes: 1—Molecular weight markers; 2, 4, 6,—HEWL lysozyme at pH 7.2, 5.5 and 9.0; 3, 5, 7—HPC extract obtained with ACN at pH 7.2, 5.5 and 9.0. Methylene blue staining.

**Figure 10 ijms-24-15607-f010:**
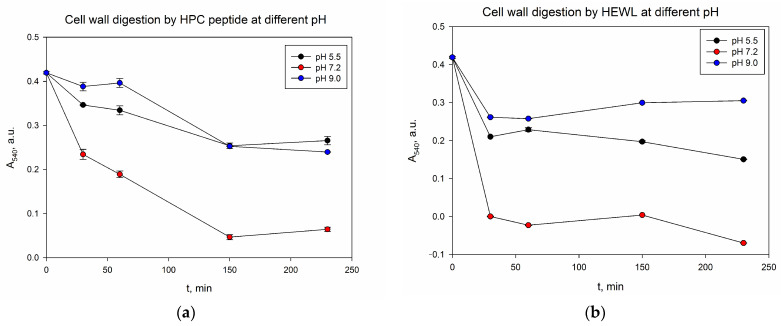
Kinetics of *M. lysodeikticus* cell wall degradation at different pH values. (**a**)—HPC extract; (**b**)—HEWL lysozyme.

**Figure 11 ijms-24-15607-f011:**
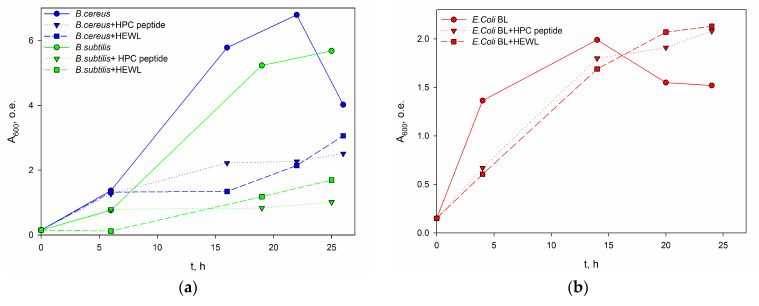
Dependence of growth of gram-positive strains of *B. cereus* and *B. subtilis* (**a**) and gram-negative strain of *E. coli* (**b**) in the presence of HPC extract and HEWL lysozyme.

## Data Availability

The data presented in this study are available upon request from the corresponding author.

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
