# Peer review of "Novel Antimicrobial Peptide from the Hepatopancreas of the Red King Crab"

_ijms, 2023, doi:10.3390/ijms242115607_

Round 1

Reviewer 1 Report

I found the manuscript really interesting . I have several issues and suggestions for authors:

1. Abstract - too complex sentence and passive (lines 17-22). 

2. Try to escape  passives along the whole text.

3. lines 427-442 - Is it the part of the Instruction for authors included mistakenly? 

4. Maybe, the authors should supplement electropherograms with additional markers to highlight the spots of interest.

5. The manuscript (English) should be thoroughly proofread and corrected.

6. Materials and methods section is hard to understand - need to restructorize. 

7. Generally, manuscript has made a good impression, has clear structure and purpose. I understand the authors present only initial stages of the research, but pure peptide extraction by HPLC/UPLC and MS analysis with primary and, maybe, secondary structure analysis are needed to be done in further studies - of course, in case, it is possible. Influence of potential reversible, by DTT, and irreversible, by formic acid, peptide modification have to be analyzed and discussed. Specific aa-seq activity and enzyme kinetics has to be analyzed. 

Moderate English editing required.

Author Response

Thank you very much for taking the time to review this manuscript and for your comments. We have tried to revise the manuscript and improve it according to your comments. 

Comments 1: Abstract - too complex sentence and passive (lines 17-22). 

Response 1: Thank you very much for your comment. We tried to improve the text

Comments 2: Try to escape passives along the whole text

Response 2: Thank you for pointing this out. We have tried to correct that.

Comments 3:  lines 427-442 - Is it the part of the Instruction for authors included mistakenly?

Response 3: My apologies, indeed, due to our carelessness during the sending process, part of the instructions got into the text. We've fixed it. Thanks for your comment.

Comments 4:  Maybe, the authors should supplement electropherograms with additional markers to highlight the spots of interest.

Response 4: We added to the electropherograms the name of the protein to which the spot belongs and an arrow to it.

Comments 5:  The manuscript (English) should be thoroughly proofread and corrected.

Response 5: We tried to improve English as much as possible.

Comments 6:  Materials and methods section is hard to understand - need to restructorize.

Response 6: Thank you very much for your comment. Indeed, after reading carefully, we realized that it was necessary to restructure this section. We tried to correct the text

Comments 7:  Generally, manuscript has made a good impression, has clear structure and purpose. I understand the authors present only initial stages of the research, but pure peptide extraction by HPLC/UPLC and MS analysis with primary and, maybe, secondary structure analysis are needed to be done in further studies - of course, in case, it is possible. Influence of potential reversible, by DTT, and irreversible, by formic acid, peptide modification have to be analyzed and discussed. Specific aa-seq activity and enzyme kinetics has to be analyzed. 

Response 7: Thank you for your kind words about our work. The next step, of course, we would like to obtain a pure preparation of the peptide, at this stage, while the review was going on, we carried out a mass spectrometric analysis and for this we first eluted the protein band with activity from the zymogram. For this sample incubation with a polysaccharide and NMR analysis were performed and obtained results. Mass spectrometry data have already been obtained, but before publication we have not yet been able to find a homologous sequence from the known databases and the transcriptome of the hepatopancreas of the king crab we have, so the work is still ongoing and, I hope, will be presented soon. Peptide isolation using DTT and formic acid was chosen when selecting a method to isolate the low molecular weight fraction. It is slightly faster than acetonitrile. Due to DTT, low molecular weight peptides go into solution faster due to the reduction of disulfide bonds, if any, while it is more difficult for high molecular weight proteins to do this, so this is additional purification However, we conducted studies on the effect of salts, as well as DTT, on the activity of the peptide in the extract. In the presence of DTT, the activity did not change.

4. Response to Comments on the Quality of English Language

Point 1: Moderate English editing required.

Response 1: We tried to improve English as much as possible.

Reviewer 2 Report

Dear Authors,

thank you for the opportunity to review the manuscript titled “Novel antimicrobial peptide from the hepatopancreas of the red king crab”. Despite the interesting research topic, the manuscript is not prepared with appropriate care.

Below are more detailed comments that I hope will help improve the quality of the manuscript

Line 69-72 - Authors should remove the text with instructions from the original form

Figure 1, 2, 3, 6 - the gels are quite blurry, in some places very difficult to interpret, they do not look like representative results that should be used for presentation in a publication

Figure 4, 5 - there is a reference in the text to "the region of bound acetyl groups and the region of amino groups", but they are not marked in any way in the NMR spectra, which would facilitate quick interpretation of the results for readers

Line 230 - missing the end of a sentence?

Figure 8 - no explanation of the reasons in the text regarding the death of E. coli cultures (control) after the 15th hour of incubation compared to samples incubated with the extract

Discussion - the results are quite poorly discussed with other literature reports, this section contains a lot of repetitions from the results and materials and methods sections. Lack of explanation and discussion of some of the results obtained.

Line 428-442 - this text should be removed

Line 443 – repetition “hepatopancreas”

Material and methods – “Testing of antibacterial activity on cell cultures” I do not understand why the authors used different OD measurement points over time for the bacterial strains tested. This was not justified or explained in any way. It would be logical to perform OD measurements at the same time for all tested samples.

In my opinion, for a broader characterization of the antimicrobial potential obtained by the authors extract, it would be worth conducting MIC and MBC analyzes for the bacterial strains they selected, which would definitely increase the scientific value of the manuscript.

Some of the names of authors of publications included in the references list are hyperlinks, please remove them.

Considering all the above-mentioned aspects, I do not recommend this manuscript for publication in International Journal of Molecular Sciences in present form. The manuscript needs general revision and then resubmission.

Author Response

Thank you very much for taking the time to review this manuscript and for your constructive remarks. We have tried to revise the manuscript and improve it according to your comments. Please find the detailed responses below.

Comments 1: Line 69-72 -Authors should remove the text with instructions from the original form

Response 1: My apologies, indeed, due to our carelessness during the sending process, part of the instructions got into the text. We've fixed it. Thanks for your comment.

Comments 2: Figure 1, 2, 3, 6 - the gels are quite blurry, in some places very difficult to interpret, they do not look like representative results that should be used for presentation in a publication

Response 2: Thank you for your comment. We have repeated the electrophoresis where possible. We marked the protein spots of interest and labeled them.

Comments 3:  Figure 4, 5 - there is a reference in the text to "the region of bound acetyl groups and the region of amino groups", but they are not marked in any way in the NMR spectra, which would facilitate quick interpretation of the results for readers

Response 3: You are right. Indeed, indicating which polysaccharide proton signals change upon incubation with the extract greatly facilitates perception. We have fixed this problem.

Comments 4:  Line 230 - missing the end of a sentence?

Response 4: Unfortunately, the design clashed with the text. We have fixed it

Comments 5:  Figure 8 - no explanation of the reasons in the text regarding the death of E. coli cultures (control) after the 15th hour of incubation compared to samples incubated with the extract

Response 5: We believe that since this is a batch bioprocess, the decrease in growth of the culture without antibacterial agents is most likely due to the fact that the culture at such a high density does not have enough nutrients, aeration is also pure, and some metabolites can inhibit growth and result in cell death. The presence of antibacterial agents prevents such a high culture density. We have given this explanation in the text of lines 477-480.

Comments 6:  Discussion - the results are quite poorly discussed with other literature reports, this section contains a lot of repetitions from the results and materials and methods sections. Lack of explanation and discussion of some of the results obtained.

Response 6: We have rewritten the discussion, taking into account the comments and wishes of the reviewers.

Comments 7:  Line 428-442 - this text should be removed

Response 7: My apologies, due to our carelessness during the sending process, part of the instructions got into the text. We've removed it.

Comments 8:  Line 443 – repetition “hepatopancreas”

Response 8: Thank you for your comment. We have removed the repeat

Comments 9:  Material and methods – “Testing of antibacterial activity on cell cultures” I do not understand why the authors used different OD measurement points over time for the bacterial strains tested. This was not justified or explained in any way. It would be logical to perform OD measurements at the same time for all tested samples.

Response 9: Thanks for the question. Originally, there were two independent experiments: B. cereus and E. coli and B. subtilis and E. coli, when the strains were provided, so the time intervals were slightly different. While writing the article, colleagues decided to separate the graphs for Bacillus and E. coli and remove one of the parallel experiments for E. coli to keep the same number of replicates.

Comments 10:  In my opinion, for a broader characterization of the antimicrobial potential obtained by the authors extract, it would be worth conducting MIC and MBC analyzes for the bacterial strains they selected, which would definitely increase the scientific value of the manuscript.

Response 10: Thank you for your comment. Although we have just started the work of isolating a pure peptide, in particular the mass spectrometry data have been obtained, so we will take this valuable note into account in the further research that is planned to be carried out on a wide range of strains from the collection of microorganisms available to our colleagues.

Comments 11:  Some of the names of authors of publications included in the references list are hyperlinks, please remove them.

Response 11: Thanks for your note, we have removed the hyperlinks

Reviewer 3 Report

The study on the novel antimicrobial peptide extracted from the hepatopancreas of the red king crab presents significant findings, although there are several notable concerns and areas for improvement.

Major Concern:

  1. The authors have discovered a peptide extract from the king crab demonstrating antimicrobial effects. However, a major concern arises from the lack of characterization of peptide purity, which is crucial for validating the results and understanding the potential applications of the peptide.

Minor Concerns:

  1. Descriptive elements specific to journal templates should be removed from the text to enhance clarity and readability.
  2. In Result 2.1, the use of SDS-PAGE could potentially denature proteins, raising questions about the positive results. The authors should provide an explanation for this phenomenon to ensure the validity of their findings.
  3. The SDS-PAGE results indicate the presence of larger proteins in the peptide extract, emphasizing the need for careful examination in all tests. Clearer photos of protein gels would enhance the presentation of these results.
  4. In Figure 2, it is essential to indicate marker bands along with their respective sizes to provide accurate context for the results.
  5. The method involving the incubation of polysaccharide with the peptide extract for five days at 37°C raises concerns about potential degradation. The authors should address how they excluded the possibility of degradation.

7.     Figure 8 reveals a drastic decrease in the growth of B. cereus after about 20 hours, requiring an explanation. Furthermore, the uncommon decrease observed in untreated bacteria samples, including E. coli, needs clarification.

8.       The study lacks a comprehensive discussion of recent literature on marine sourced antimicrobial peptides. Relevant publications, such as those in Front Mar Sci. 2023;9:2880. doi:10.3389/FMARS.2022.1112595; Nat. Prod. Rep. 2023. DOI https://doi.org/10.1039/D3NP00031A; Front Microbiol. 2021;12. doi:10.3389/FMICB.2021.785085, should be incorporated into the discussion.

See comments

Author Response

Thank you very much for taking the time to review this manuscript and for your constructive remarks. We have tried to revise the manuscript and improve it according to your comments. 

Comments 1: The authors have discovered a peptide extract from the king crab demonstrating antimicrobial effects. However, a major concern arises from the lack of characterization of peptide purity, which is crucial for validating the results and understanding the potential applications of the peptide.

Response 1: Thank you for your comment. Although we have just started the work of isolating a pure peptide, in particular the mass spectrometry data have been obtained. For this we first eluted the protein band with activity from the zymogram. For this sample incubation with a polysaccharide and NMR analysis were performed and obtained results were added to paper.

Comments 2: Descriptive elements specific to journal templates should be removed from the text to enhance clarity and readability.

Response 2: My apologies, indeed, due to our carelessness during the sending process, part of the instructions got into the text. We've fixed it. Thanks for your comment.

Comments 3:  In Result 2.1, the use of SDS-PAGE could potentially denature proteins, raising questions about the positive results. The authors should provide an explanation for this phenomenon to ensure the validity of their findings

Response 3: To detect activity, we used the zymography method. After the end of gel electrophoresis, the gel is washed from SDS excess and kept in renaturation buffer with Triton X-100 for 2 hours at 37°C with shaking. Already after this time, after staining, protein bands can be seen, which indicates its renaturation and restoring of activity. We have added this to 2.1.

Comments 4:  The SDS-PAGE results indicate the presence of larger proteins in the peptide extract, emphasizing the need for careful examination in all tests. Clearer photos of protein gels would enhance the presentation of these results

Response 4: Thank you for your comment. We have repeated the electrophoresis where possible. We marked the protein spots of interest and labeled them.

Comments 5:  In Figure 2, it is essential to indicate marker bands along with their respective sizes to provide accurate context for the results.

Response 5: Thank you for the comment, we have put the risks against the markers

Comments 6:  The method involving the incubation of polysaccharide with the peptide extract for five days at 37°C raises concerns about potential degradation. The authors should address how they excluded the possibility of degradation.

Response 6: Thank you for your valuable comments. During the incubation, we included all controls, both the polysaccharide and the peptide extract separately, as well as the reaction mixture. NMR spectra were recorded for all samples to analyze degradation, including the polysaccharide before incubation, which we have added to the article.

Comments 7:  Figure 8 reveals a drastic decrease in the growth of B. cereus after about 20 hours, requiring an explanation. Furthermore, the uncommon decrease observed in untreated bacteria samples, including E. coli, needs clarification.

Response 7: We believe that since this is a batch bioprocess, the decrease in growth of the culture without antibacterial agents is most likely due to the fact that the culture at such a high density does not have enough nutrients, aeration is also pure, and some metabolites can inhibit growth and result in cell death. The presence of antibacterial agents prevents such a high culture density. We have given this explanation in the text of lines 477-480.

Comments 8:  The study lacks a comprehensive discussion of recent literature on marine sourced antimicrobial peptides. Relevant publications, such as those in Front Mar Sci. 2023;9:2880. doi:10.3389/FMARS.2022.1112595; Nat. Prod. Rep. 2023. DOI https://doi.org/10.1039/D3NP00031A; Front Microbiol. 2021;12. doi:10.3389/FMICB.2021.785085, should be incorporated into the discussion

Response 8: Thank you for the valuable suggestion. We have rewritten the discussion and added the suggested recent publications to the paper.

4. Response to Comments on the Quality of English Language

Point 1 Minor editing of English language required

Response 1: We tried to improve English as much as possible.

Round 2

Reviewer 3 Report

The authors have addressed the previously raised comments.